# The Research on Anti-Nickel Contamination Mechanism and Performance for Boron-Modified FCC Catalyst

**DOI:** 10.3390/ma15207220

**Published:** 2022-10-17

**Authors:** Chengyuan Yuan, Lei Zhou, Qiang Chen, Chengzhuang Su, Zhongfu Li, Guannan Ju

**Affiliations:** 1School of Materials Science and Engineering, Shandong University of Technology, Zibo 255000, China; 2Shanxi Tengmao Technology Co., Ltd., Hejin 043300, China

**Keywords:** anti-nickel contamination, boron modification, FCC catalyst, mechanism, performance

## Abstract

Fluid catalytic cracking (FCC) is still a key process in the modern refining area, in which nickel-contamination for an FCC catalyst could obviously increase the dry gas and coke yields and thus seriously affect the stability of the FCC unit. From the points of surface acidity modification and Ni-passivation, in this paper, a boron-modified FCC catalyst (BM-Cat) was prepared using the in situ addition method with B_2_O_3_ as a boron source and emphatically investigated its mechanism and performance of anti-nickel contamination. The mechanism research results suggested that, in calcination, boron could destroy the structure of the Y zeolite and thus decrease the total acid sites and strong acid sites of the Y zeolite from 291.5 and 44.6 μmol·g^−1^ to 244.2 and 32.1 μmol·g^−1^, respectively, which could obviously improve the dry gas and coke selectivity of the catalyst and thus enhance the nickel capacity for BM-Cat; on the other hand, under hydrothermal conditions, boron could react with NiO and form into NiB_2_O_4_, which could obviously raise the range of the reduction temperature for NiO from 350–600 °C to 650–800 °C and thus promote the nickel-passivation ability for BM-Cat. Therefore, evaluation results of heavy oil catalytic cracking indicated that, under the same nickel-contamination condition, in contrast to the compared catalyst (C-Cat), the dry gas yield, coke yield, and H_2_/CH_4_ of BM-Cat obviously decreased by 0.77 percentage points, 2.09 percentage points, and 13.53%, respectively, with light yield and total liquid yield increasing by 3.25 and 2.08 percentage points, respectively, which fully demonstrates the excellent anti-nickel contamination performance of BM-Cat.

## 1. Introduction

Presently, fluid catalytic cracking (FCC) is still a main processing technology for crude oil lightening and has occupied a very important position in the refining area, in which the FCC catalyst is considered as a critical factor that could greatly influence the state of the FCC unit [1,2,3]. In recent years, as the quality deterioration of crude oil has become more and more serious worldwide, the heavy metals content in crude oil has been significantly increased, which has called for the excellent heavy metals-tolerance performance of FCC catalysts [4,5,6].

Generally, V, Ni, and Fe are the main contamination species and have different contamination mechanisms for FCC catalysts [7]. Among them, nickel contamination for FCC catalysts could lead to the obvious increasing of dry gas and coke yields, which could seriously influence the running stability of the FCC unit [8]. The nickel contamination mechanism is that, during the regeneration of the FCC catalyst, the nickel species in the feed oil first deposit on the surface of the FCC catalyst as NiO. Then, in the reduced circumstance of the FCC reaction, the above NiO could be reduced into Ni^0^ species that have high dehydrogenation activity, which could obviously promote the dehydrogenation reactions for feed oil molecules and thus cause the increasing of dry gas and coke yields [9,10]. To improve the anti-nickel contamination performance of FCC catalysts, some special types of materials such as macroporous and mesoporous alumina are usually used as matrix components to decrease the dry gas and coke yields by improving the diffusion properties of FCC catalysts [11,12]. However, these special matrix materials not only could influence the physiochemical properties of the prepared FCC catalyst but also obviously increase the cost of FCC catalysts, which make them difficult to be used in practice [13,14]. According to the above Ni-contamination mechanism, tow aspects must be considered to improve the anti-nickel contamination performance of FCC catalysts. One aspect, the surface acidity of FCC catalysts, should be appropriately modified to decrease the yields of dry gas and coke by restraining the over-cracking and coking reactions. FCC catalysts should also possess excellent Ni-passivation ability to prevent NiO from being reduced into Ni^0^ species and thus decrease the dehydrogenation reactions caused by Ni^0^ species.

Therefore, in this study, for the further development of anti-nickel contaminated FCC catalysts with practical application prospects, a boron-modified FCC catalyst was prepared using the in situ modification method using cheap and easily-obtained B_2_O_3_ as a boron source. Emphatically, the anti-nickel contamination mechanism and performance of boron-modified FCC catalysts were investigated by X-ray diffraction (XRD), NH_3_ temperature programed desorption (NH_3_-TPD), pyridine-adsorption Fourier transform infrared spectra (Py-FTIR), H_2_ temperature programed reduction (H_2_-TPR), and advanced cracking equipment (ACE). ACE results indicate that the boron-modified FCC catalyst exhibited excellent anti-nickel contamination performance in contrast to the compared FCC catalyst. Additionally, the advantages of the easy production process and low cost would make good application prospects for the boron-modified FCC catalysts developed in this work.

## 2. Materials and Methods

### 2.1. Materials

A REUSY molecular sieve, alumina sol, and kaolin were provided by Lanzhou Petrochemical Company, industrial grade. B_2_O_3_ and Ni(NO_3_)_2_·6H_2_O were purchased from Sinopharm Chemical Reagent Company, analytically pure.

### 2.2. Preparation of FCC Catalysts

The FCC catalysts were prepared following to the semi-synthesis method [15]. The detailed processes are below:

Compared FCC catalyst: A measured REUSY molecular sieve, alumina sol, kaolin, and deionized water were thoroughly mixed, and then the above slurry was spray-dried for shaping. After that, the above-obtained solid microspheres were calcinated (450 °C, 30 min), washed, and dried in sequence to produce the FCC catalyst (C-Cat). The components’ mass ratio for C-Cat was REUSY:alumina sol (based on Al_2_O_3_):kaolin = 35:10:55.

Boron-modified FCC catalyst: A measured REUSY molecular sieve, alumina sol, kaolin, B_2_O_3_, and deionized water were thoroughly mixed, and then the above slurry was spray-dried for shaping. After that, the above-obtained solid microspheres were calcinated (450 °C, 30 min), washed, and dried in sequence to produce the boron-modified FCC catalyst (B-Cat). The components’ mass ratio for B-Cat was REUSY:alumina sol (based on Al_2_O_3_):(kaolin+B_2_O_3_) = 35:10:55.

### 2.3. Characterizations and Evaluations

X-ray diffraction (XRD) was carried out on a PANalytical X’pert pro diffractometer (Malvern, UK), operating at 40 kV, 40 mA, and scanning from 5° to 75° at a speed of 0.01°/s. Pyridine-adsorption Fourier transform infrared spectra (Py-FTIR) were recorded on a Burker TENSOR 27 instrument (Billerica, MA, USA). All samples were activated at 300 °C for 3 h before pyridine adsorption. NH_3_ temperature programed desorption (NH_3_-TPD) was performed on a Micromeritics AUTOCHEM II 2920 chemisorption instrument (Micromeritics, Norcross, GA, USA) in the range of 100–500 °C at a heating rate of 15 °C/min. The adsorption of ammonia on the samples was performed at room temperature, followed by removing physically adsorbed ammonia at 100 °C for 1 h in flowing pure nitrogen. H_2_ temperature-programmed reduction (H_2_-TPR) was measured on a Micromeritics AUTOCHEM II 2920 chemisorption instrument.

The prepared FCC catalyst was contaminated by nickel (Ni: 7000 ppm) using the incipient-wetness impregnation method with Ni(NO_3_)_2_·6H_2_O as a nickel source. The heavy oil catalytic cracking performance of the FCC catalysts after aging treatment (800 °C, 17 h in 100% steam) was evaluated on an advanced cracking evaluation unit (ACE, Kayser. R+Multi). The properties of the heavy oil feed are listed in Table 1.

## 3. Results

### 3.1. Analysis of Anti-Nickel Contamination Mechanism for Boron-Modified FCC Catalyst

#### 3.1.1. The Enhancing of Nickel-Capacity for FCC Catalyst by Boron Modification

Y zeolite is a critical activity component of FCC catalysts and provides most of the surface acid sites for FCC catalysts [16]. Therefore, to investigate the influence of boron modification on the physicochemical properties for the REUSY zeolite, REUSY was modified by B_2_O_3_ using the incipient-wetness impregnation method. After calcination at 540 °C for 3 h, the boron-modified REUSY (B-Y) was obtained. Figure 1 shows the XRD patterns of Y zeolites; as is shown in Figure 1a, compared with the REUSY parent, B-Y still displayed the diffraction characteristics of Y zeolite, which indicated that B-Y still generally kept the crystalline phase of the Y zeolite after boron modification [17]. The diffraction intensity of B-Y was distinctly lower than that of REUSY, which suggests that the crystallinity of B-Y reduced to some extent because of boron modification [18]. For further comparison, the diffraction of the (533) lattice plane was magnified because the diffraction intensity of the (533) lattice plane is commonly used to determine the crystallinity of Y zeolite [19]. As is displayed in Figure 1b, the diffraction intensity of the (533) lattice plane for B-Y was evidently decreased in contrast to REUSY, with the diffraction peak shifting toward a large angle, suggesting that the structure of B-Y was destroyed with decreasing of the crystal cell size due to boron modification, which is in good accordance with the results reported by Feng et al. [20].

Figure 2 shows the NH_3_-TPD profiles of REUSY and B-Y. It is can be seen that both REUSY and B-Y exhibited two obvious NH_3_-desorption peaks at 100–300 °C and 300–500 °C, respectively, which could be attributed to weak acid sites and strong acid sites, respectively [21]. Because of the structure destruction caused by boron modification, compared with the REUSY parent, the total area of the above two NH_3_-desorption peaks of B-Y (especially the peak for strong acid sites) decreased, which suggests that the quantities of total acid sites and strong acid sites for B-Y was increased by boron modification.

The Py-FTIR spectra of REUSY and B-Y are shown in Figure 3. As is shown, both REUSY and B-Y exhibited IR bands at around of 1450 and 1540 cm^−1^, which could be attributed to L acid sites and B acid sites, respectively [22]. Compared with REUSY, the areas of the above IR bands for B-Y decreased, indicating the reducing of acid sites’ quantity for B-Y.

The crystallinities and detailed acid sites quantities calculated from Py-FTIR of REUSY and B-Y are listed in Table 2. As is listed, compared with REUSY, the crystallinity of B-Y reduced by three percentage points. The total acid sites quantity of B-Y decreased from 291.5 to 244.2 μmol·g^−1^ with the strong acid sites quantity of B-Y decreasing from 44.6 to 32.1 μmol·g^−1^ in contrast to REUSY.

The NH_3_-TPD profiles of the prepared FCC catalysts are shown in Figure 4. As is shown, compared with C-Cat, the areas of the total NH_3_-desorption peak and the peak at about 350–450 °C for BM-Cat decreased distinctly, indicating that the quantities of total acid sites and strong acid sites for BM-Cat were decreased, which could be attributed to the above acidity adjustment by boron modification to the REUSY zeolite. The detailed acid properties of the FCC catalysts are listed in Table 3. As is exhibited, compared with C-Cat, the total acid sites quantity for BM-Cat decreased from 223.6 to 196.9 μmol·g^−1^, with strong acid sites’ quantity decreasing from 29.4 to 21.8 μmol·g^−1^. It has been reported that appropriate decreasing of acid sites (especially for strong acid sites) could effectively improve the dry gas and coke selectivities for FCC catalysts by restraining the over-cracking and coking reactions during the FCC process [23], which would make good dry gas and coke selectivities for BM-Cat due to the boron modification.

The heavy oil catalytic cracking performances for fresh FCC catalysts are shown in Table 4. It can be seen that, compared with C-Cat, the dry gas and coke yields of BM-Cat obviously decreased by 0.19 and 1.43 percentage points, respectively, with light and total liquid yields increasing by 1.98 and 0.97 percentage points, respectively, which suggests that the dry gas and coke selectivities for BM-Cat was effectively improved by boron modification. For a FCC catalyst, good dry gas and coke selectivities would be very helpful for enhancing the nickel capacity of the catalyst through offsetting the increasing of dry gas and coke yields that are caused by nickel contamination, and thus could improve the anti-nickel contamination performance of BM-Cat.

#### 3.1.2. The Promotion of Nickel-Passivation Ability for FCC Catalysts by Boron Modification

Based on the abovementioned nickel-contamination mechanism for FCC catalysts, it could be concluded that the anti-nickel contamination performance of FCC catalysts would be improved by preventing the reduction of NiO (nickel-passivation) during FCC reaction [24]. Figure 5 shows the H_2_-TPR profiles of nickel-contaminated FCC catalysts. As is shown, the reduction temperature range of NiO species on C-Cat was about 350–500 °C. By contrast, due to boron modification, the reduction temperature range of NiO species on BM-Cat was obviously raised to 650–800 °C, which was much higher than the temperature of traditional FCC reaction, which suggests that the reduction of NiO species on BM-Cat would be largely prevented by boron modification during FCC reaction. Therefore, the above results demonstrate that the nickel-passivation ability of FCC catalysts could evidently be promoted by boron modification, which would be beneficial for the improvement of the anti-nickel contamination performance of FCC catalysts.

To investigate the reason of the nickel-passivation ability promotion of BM-Cat by boron modification, as shown in Figure 1, two model reactions were made under the hydrothermal condition that is similar to the aging treatment of FCC catalysts.

The XRD patterns of the products for model reactions are shown in Figure 6. As is shown, product 1 exhibited diffraction peaks at around 37°, 43°, and 63°, which could be attributed to the characteristic diffraction of NiO. Product 2 displayed diffraction peaks at around 23°, 26°, 34°, 36°, 40°, 42°, 51°, 53°, 55°, 60°, and 61°, which could be attributed to the characteristic diffraction of NiB_2_O_4_ and indicates that NiO species and B_2_O_3_ could react and form into NiB_2_O_4_ under hydrothermal conditions [25].

The H_2_-TPR profiles of NiO and NiB_2_O_4_ are displayed in Figure 7. As is exhibited, the reduction temperature of pure NiO was in the range of 350–600 °C. Compared to pure NiO, the reduction temperature of NiO in NiB_2_O_4_ was obviously raised to 650–800 °C, which suggests that the reduction of NiO in NiB_2_O_4_ was much more difficult than that of pure NiO. Therefore, based on the above results, it could be deduced that this was probably because of the formation of NiB_2_O_4_ that caused the excellent nickel-passivation ability of BM-Cat in contrast to C-Cat.

### 3.2. Heavy Oil Catalytic Cracking Performance of Nickel-Contaminated FCC Catalysts

The heavy oil catalytic cracking performances of nickel-contaminated FCC catalysts are listed in Table 5. As is listed, under the same nickel-contamination condition, in comparison with C-Cat, the dry gas and coke yields of BM-Cat significantly decreased by 0.77 and 2.09 percentage points, respectively, with light yield and total liquid increasing by 3.25 and 2.08 percentage points, respectively, because of the suitable surface acidity and excellent Ni-passivation ability of BM-Cat. Moreover, the H_2_/CH_4_ value for BM-Cat was obviously decreased by 13.53% in contrast to C-Cat, suggesting that the dehydrogenation activity of BM-Cat had been effectively suppressed. Therefore, in summary, the above heavy oil catalytic cracking results definitely demonstrated that the anti-nickel contamination performance of FCC catalysts could be greatly improved by boron modification.

## 4. Conclusions

(1) Boron-modified FCC catalysts were prepared using the in situ addition method with B_2_O_3_ as a boron source, and its mechanism and performance were investigated for anti-nickel contamination in heavy oil catalytic cracking through NH_3_-TPD, Py-FTIR, H_2_-TRP, and ACE unit.

(2) The examination of the anti-nickel contamination mechanism indicated that the quantities of total acid sites and strong acid sites for boron-modified FCC catalysts were appropriately decreased because of the destruction of boron modification on the Y zeolite, which was greatly helpful for the improvement of dry gas and coke selectivity and thus could obviously enhance the nickel capacity of FCC catalysts; on the other hand, B_2_O_3_ and NiO could react and form into NiB_2_O_4_ under hydrothermal conditions and thus obviously raised the reduction temperature of NiO, which could greatly restrict the reduction reaction of NiO species and thus promote the nickel-passivaiton ability of boron-modified FCC catalysts. Because of the above two reasons, the boron-modified FCC catalysts exhibited significant anti-nickel contamination performance in contrast to the compared FCC catalyst.

(3) The research content could serve as a foundation for further development of new anti-nickel contaminated FCC catalysts. Meanwhile, the boron-modified FCC catalyst in this paper would have good application prospects because of its obvious advantages, such as high performance, easy production process, and low cost.

## Data Availability

Not applicable.

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
