# Peer review of "The Research on Anti-Nickel Contamination Mechanism and Performance for Boron-Modified FCC Catalyst"

_materials, 2022, doi:10.3390/ma15207220_

Round 1
Reviewer 1 Report
I recommend the publication of this paper in the Materials after a major revision:
1- What is the novelty of this work? Novelty should be highlighted.
2- In the Abstract, the first paragraph (line 7-9) should be deleted and moved to the Introduction.
3- Some new relevant references (from 2020, 2021, and 2022) should be added to the text.
Author Response
Dear Ms. Haney Liu,
Thank you for your feedback regarding our manuscript entitled “The Research on Anti-Nickel Contamination Mechanism and Performance for Boron-Modified FCC Catalyst” (materials- 1908337). We appreciate the constructive and insightful critiques provided by the reviewers. We have made extensive efforts to further strengthen the manuscript by making focused revisions to improve the clarity of our explanation and presentation of our experimental approaches and goals in order to address all of the reviewers’ concerns. We have addressed all the technical concerns raised by the reviewers through major revisions. Meanwhile, Qiang Chen performs the test of Figure 3 in this revised manuscript. We could consider him as the third author, with the consent of the editor.
We remain highly appreciative of your time and consideration. We look forward to hearing from you.
Best regards,
Yours sincerely,
Guannan Ju
ADDENDUM – POINT-BY-POINT RESPONSE TO REVIEWER’ COMMENTS
We would like to thank the reviewers for taking their time to review our manuscript. Their comments have helped us strengthen this paper and revisions have been made to address the reviewers’ concerns as well as improve on the clarity and focus of the manuscript. Highlighted in yellow in the manuscript are the revisions that have been made.
Reviewer 1:
I recommend the publication of this paper in the Materials after a major revision:
1 What is the novelty of this work? Novelty should be highlighted.
Reply: Because the research on anti-nickel contamination of FCC catalyst has not been paid much attention. There is not an effective way to obviously improve the anti-nickel contamination performance of FCC catalyst in practice. Based on this fact, our work provided a simple, cheap and highly-effective way for the preparation of FCC catalyst that with excellent anti-nickel contamination performance. Moreover, the research of anti-nickel contamination mechanism of boron-modification on FCC catalyst in this work also would provide a valuable theory basis for the development of new-type anti-nickel contaminated FCC catalyst.
2- In the Abstract, the first paragraph (line 7-9) should be deleted and moved to the Introduction.
Reply: We have made corresponding corrections in revised manuscript.
3- Some new relevant references (from 2020, 2021, and 2022) should be added to the text.
Reply: We have added related references in revised manuscript.

Reviewer 2 Report
I believe a promising study was done on anti-nickel contamination and the research could be published after minor revision.
1- Introduction is very brief.
2- please add reference for FCC preparation.
3- The instrumentation in the material and methods section is not complete. for example for the XRD, the analysis conditions should be added like the lamp, wavenumber, scan rate, ...
4- In the result section, please added miller indices to the figures 1, and 5.
5- A detailed comparison should be done with the previous strategies for preventing nickel contamination.
6- The references should be developed with more focus on the FCC catalyst.
7- please add references about heterogeneous catalysts like: a) Applied Organometallic Chemistry 2017, 31 (11), e3774; b)RSC Advances 2015, 5 (59), 47617-47620; c) Cellulose 2020, 27 (2), 895-904.
Author Response
Dear Ms. Haney Liu,
Thank you for your feedback regarding our manuscript entitled “The Research on Anti-Nickel Contamination Mechanism and Performance for Boron-Modified FCC Catalyst” (materials- 1908337). We appreciate the constructive and insightful critiques provided by the reviewers. We have made extensive efforts to further strengthen the manuscript by making focused revisions to improve the clarity of our explanation and presentation of our experimental approaches and goals in order to address all of the reviewers’ concerns. We have addressed all the technical concerns raised by the reviewers through major revisions. Meanwhile, Qiang Chen performs the test of Figure 3 in this revised manuscript. We could consider him as the third author, with the consent of the editor.
We remain highly appreciative of your time and consideration. We look forward to hearing from you.
Best regards,
Yours sincerely,
Guannan Ju
ADDENDUM – POINT-BY-POINT RESPONSE TO REVIEWER’ COMMENTS
We would like to thank the reviewers for taking their time to review our manuscript. Their comments have helped us strengthen this paper and revisions have been made to address the reviewers’ concerns as well as improve on the clarity and focus of the manuscript. Highlighted in yellow in the manuscript are the revisions that have been made.
Reviewer 2:
I believe a promising study was done on anti-nickel contamination and the research could be published after minor revision.
1- Introduction is very brief.
Reply: We have added supplemental contents into the introduction of revised manuscript.
2- please add reference for FCC preparation.
Reply: We have added related reference in the section of FCC preparation in revised manuscript.
3- The instrumentation in the material and methods section is not complete. for example for the XRD, the analysis conditions should be added like the lamp, wavenumber, scan rate, ...
Reply: We have added the related contents in revised manuscript.
4- In the result section, please added miller indices to the figures 1, and 5.
Reply: We have made corrections in revised manuscript.
5- A detailed comparison should be done with the previous strategies for preventing nickel contamination.
Reply: Thank you very much for your constructive advice. The macroporous and mesoporous aluminas are being prepared according to the previous reports. The application of these materials in anti-nickel contamination of FCC catalyst would be next work for us.
6- The references should be developed with more focus on the FCC catalyst.
Reply: We have made corresponding adjustments on references in revised manuscript.
7- please add references about heterogeneous catalysts like: a) Applied Organometallic Chemistry 2017, 31 (11), e3774; b) RSC Advances 2015, 5 (59), 47617-47620; c) Cellulose 2020, 27 (2), 895-904.
Reply: We have added related contents in revised manuscript.

Reviewer 3 Report
Kindly refer the comments.

Author Response
Dear Ms. Haney Liu,
Thank you for your feedback regarding our manuscript entitled “The Research on Anti-Nickel Contamination Mechanism and Performance for Boron-Modified FCC Catalyst” (materials- 1908337). We appreciate the constructive and insightful critiques provided by the reviewers. We have made extensive efforts to further strengthen the manuscript by making focused revisions to improve the clarity of our explanation and presentation of our experimental approaches and goals in order to address all of the reviewers’ concerns. We have addressed all the technical concerns raised by the reviewers through major revisions. Meanwhile, Qiang Chen performs the test of Figure 3 in this revised manuscript. We could consider him as the third author, with the consent of the editor.
We remain highly appreciative of your time and consideration. We look forward to hearing from you.
Best regards,
Yours sincerely,
Guannan Ju
ADDENDUM – POINT-BY-POINT RESPONSE TO REVIEWER’ COMMENTS
We would like to thank the reviewers for taking their time to review our manuscript. Their comments have helped us strengthen this paper and revisions have been made to address the reviewers’ concerns as well as improve on the clarity and focus of the manuscript. Highlighted in yellow in the manuscript are the revisions that have been made.
Reviewer 3:
I came across several discrepancies in the manuscript. Thereby, the manuscript is subjected for
MAJOR REVISION. Some of my observations are as follows;
- The writing needs to be improved throughout the manuscript. There are many
grammatical, spelling and tenses errors as well.
Reply: We have checked and corrected the related errors in revised manuscript.
- Abstract:
Abstract should consist of 70-90% of results and discussion. Briefly highlight on the
characterization results as well. Provide quantitative characterization results.
Reply: We have added supplemental data in the abstract of revised manuscript.
- Methodology
Explain in detailed the preparation of FCC catalyst (the amount, period, temperature
etc).
Briefly explain the experimental condition required for XRD, TPD and other
instrumentation. How the author carried out FTIR Pyridine analysis?
Reply: We have added related contents in revised manuscript.
- Results & Discussion:
Figure 1, page 3, provide the caption of y-axis.
Page 4, Line 110, typo error NH3.
Page 10 , Line 174, typo error NiB2O4.
Where is the figure for pyridine FTIR?
For each of the figures and tables, provide relevant discussion rather than interpreting
the data only. No detailed correlation with the characterization results was discussed.
Why the author never compares their present work with other published works?
Reply: We have made corresponding corrections in revised manuscript. The figures for pyridine FTIR have been added in revised manuscript.
- References
Include recent publications, rather than citing publication of 1968, 1988,1991, 1997-
- This proves the introduction or discussion are not up-to-date and are lacking of
important discussions and reviews.
Reply: We have added latest references in revised manuscript.

Round 2
Reviewer 1 Report
The work was considerably improved, and I suggest to accept the manuscript in its current form for publication in the Materials.
Author Response
Dear editor,
Thank you for your feedback regarding our manuscript entitled “The Research on Anti-Nickel Contamination Mechanism and Performance for Boron-Modified FCC Catalyst” (materials- 1908337). We have addressed all the technical concerns raised by the reviewers through minor revisions.
We remain highly appreciative of your time and consideration. We look forward to hearing from you.
Best regards,
Yours sincerely,
Guannan Ju
Reviewer 3 Report
Kindly proof-read the manuscript and ensure it is free from errors
Relate the characterization and catalysis and briefly explain on it.
Author Response
Dear editor,
Thank you for your feedback regarding our manuscript entitled “The Research on Anti-Nickel Contamination Mechanism and Performance for Boron-Modified FCC Catalyst” (materials- 1908337). We have addressed all the technical concerns raised by the reviewers through minor revisions.
We remain highly appreciative of your time and consideration. We look forward to hearing from you.
Best regards,
Yours sincerely,
Guannan Ju
ADDENDUM – POINT-BY-POINT RESPONSE TO REVIEWER’ COMMENTS
We would like to thank the reviewers for taking their time to review our manuscript. Their comments have helped us strengthen this paper and revisions have been made to address the reviewers’ concerns as well as improve on the clarity and focus of the manuscript. Highlighted in green in the manuscript are the revisions that have been made.
Reviewer 3:
Kindly proof-read the manuscript and ensure it is free from errors
Relate the characterization and catalysis and briefly explain on it.
.
Reply: We have revised the relation the characterization and catalysis in revised manuscript.
